# Evaluation of Malignancy Risk in 18F-FDG PET/CT Thyroid Incidentalomas

**DOI:** 10.3390/diagnostics9030092

**Published:** 2019-08-07

**Authors:** Maria-Iulia Larg, Dragoș Apostu, Claudiu Peștean, Katalin Gabora, Iulian Claudiu Bădulescu, Elena Olariu, Doina Piciu

**Affiliations:** 1Prof. Dr. Ion Chiricuță Institute of Oncology, Department of Nuclear Medicine, 400015 Cluj-Napoca, Romania; 2Faculty of Medicine, Iuliu Hațieganu University of Medicine and Pharmacy, 400012 Cluj-Napoca, Romania

**Keywords:** thyroid incidentalomas, FDG PET/CT, malignancy risk

## Abstract

Thyroid incidentalomas detected by 18 fluoro-2-deoxy-d-glucose (FDG)-positron emission tomography/computed tomography (PET/CT) are a real challenge for nuclear medicine physicians and clinicians. This study aimed to evaluate the risk of malignancy for patients with focal thyroid incidentalomas (TIs) diagnosed through FDG PET/CT. Data from 6900 patients, with a known primary tumor, who had an FDG PET/CT investigation performed were analyzed for the presence of incidental thyroid uptake. The focal TIs were reported, and the patients were referred for further investigation to the endocrinology department. There were 126 patients (1.82%) who presented with focal thyroid uptake, and for 87 of them, investigations were completed with ultrasonography (US), and for 29 with a fine needle aspiration biopsy (FNAB) procedure. Malignancy was detected in 7.93% (10/126) of cases. An arbitrary cutoff value of four was established for the standard uptake value lean body mass (SUVlbm Max) to differentiate the malignant nodules from the benign ones, and this value was significantly associated with malignancy (*p* = 0.0168). TIs are not so frequent, but they have a potential malignancy risk, and a proper evaluation is required. Even though SUVlbm Max is a predictive factor for malignancy, the FNAB remains the main diagnostic method for the therapeutic management of these patients.

## 1. Introduction

Thyroid incidentalomas (TIs) are one of the most common incidental findings on imaging studies. A TI is defined as an unexpected and asymptomatic thyroid nodule discovered accidentally during imaging studies or a surgical intervention performed for unrelated pathologies of the thyroid [1,2].

It has been observed that thyroid nodules are common in clinical practice. Their prevalence largely depends on the method of screening and the population evaluated. Increasing age, female gender, iodine deficiency, and a history of head and neck radiation seem to increase the risk of thyroid nodules [3,4]. Also, it is estimated that clinically unsuspected nodules have been discovered in 50–60% of patients during autopsy [1,3]. 

The advances in technology, especially improvements in the quality of images in ultrasonography (US), computed tomography (CT), magnetic resonance imaging (MRI), and positron emission tomography/computed tomography (PET/CT) with different tracers such as 18 fluoro-2-deoxy-d-glucose (FDG), radiolabeled prostate-specific membrane antigen (PSMA), Fluorine-18 (F18), or Carbon-11 (C11) have considerably increased the rate detection of thyroid incidentalomas and brought multiple benefits regarding the diagnostic approach [4,5,6,7].

FDG PET/CT is a nuclear medicine technique that is based on glucose metabolism from malignant cells becoming a gold standard, especially in oncology. Even if it is not a primary investigation method in thyroid pathology, its usefulness to detect the recurrent disease during follow-up programs for patients with differentiated thyroid carcinoma is well established [8,9]. The extensive use of FDG PET/CT leads to an increasing number of discovered thyroid incidentalomas [10]. The normal thyroid gland shows a very low-grade of FDG uptake, and usually, it is slightly visualized on PET/CT scans. An incidental FDG high-uptake in the thyroid parenchyma can be diffuse, which is most often represented by an inflammatory thyroid disease like thyroiditis or focal uptake, represented by thyroid nodules [1]. The literature presents a malignancy rate in TIs that ranges from 10% to 64%, but in all of the studies, the primary limitation is the small subsets of patients further biopsied, most of them being just followed further [11]. 

The clinical implications of TIs and the impact on the therapeutic management for the primary tumor should be considered, but a main question arises, namely: is it better to continue the treatment for the primary tumor or to start the diagnosis for TIs?

At this moment, in our medical institution, there is no standardized protocol for TIs detected through FDG PET/CT. This study aims to assess the rate of malignancy in patients with focal FDG uptake of oncology patients, to adopt a standardized strategy and protocols for TIs by improving the quality of care for our patients.

## 2. Materials and Methods 

This study was approved by the ethics committee of our institution, and all of the patients included in the study signed an informed consent. 

The files from 6900 patients who underwent an FDG PET/CT scan between June 2013 to August 2018 for cancer evaluation (staging, monitoring of the disease, or treatment response) were prospectively analyzed. Incidental findings in the thyroid gland based on CT and fused PET/CT images were recorded, and 534 patients with thyroid findings were identified. A number of 256 patients out of 6900 scans (3.71%) presented just morphological CT thyroid modifications without metabolic FDG uptake; 152 patients (2.2%) were diagnosed with increased diffuse thyroid uptake; and 126 patients (1.82%) presented focal uptake, which represents our group of interest. Patients with known thyroid carcinoma and other thyroid pathologies, or patients who refused to perform supplementary thyroid investigations because of their medical condition, based on the primary cancer, were excluded from the study group. The type of primary malignant diseases is presented in Figure 1.

### F18-FDG PET/CT Imaging Protocol

After a fasting period of 4–6 h and proper hydration, the blood glucose level was analyzed (all of the patients had a serum glucose value <150 mg/dL). Administrated doses of tracer were calculated according to the patients’ weight and internal department protocol (185–481 MBq), and were intravenously administrated. FDG PET/CT scans from the vertex to proximal femur were performed on a GE Optima 560 PET/CT with a bismuth germanium oxide (BGO) detector and 16 slices of CT. The CT non-contrast images were acquired with a low-dose protocol (100 kV and 50–100 mA) for attenuation correction and to localize lesions. The PET images were obtained after 60 ± 10 min of uptake time with 2.5 min/bed, and were reconstructed through the ordered subsets expectation maximization (OSEM) iterative technique with two iterations and 16 subsets.

The images were analyzed by certified nuclear medicine physicians and radiologists. The standardized uptake value lean body mass (SUVlbm Max) was used as a semi-quantitative parameter for 18F-FDG uptake calculation, respecting a standard protocol on the workstation (Volumetrix for PET/CT) using a spherical region of interest (ROI).

Incidental thyroid increased uptake was evaluated and divided into a diffuse or focal uptake. TIs were reported as an increased focal FDG uptake on a PET/CT scan, higher than the rest of the thyroid parenchyma uptake and above the liver uptake. After the PET/CT scan, the patients with a focal FDG uptake were referred to the endocrinology department for clinical evaluation and further investigations, namely: laboratory tests and thyroid ultrasound (US) and fine-needle aspiration biopsy (FNAB), according to the guidelines recommendations. 

The identified nodules were described, and information like the number, size, margins, echogenicity, their vascularity, and the presence of calcification was reported. FNAB was recommended for all of the nodules higher than 1 cm and those with high-risk ultrasonography patterns. The cytopathological results were reported according to the Bethesda classification system [12,13]. 

Statistical analysis was performed using GraphPad Prism 6.0 software. The normal distribution was calculated using the Shapiro–Wilk test. Student t-test for unequal variances was used to compare the SUVlbm Max mean between the benign and malignant lesions. *p*-Values less than 0.05 were considered statistically significant. Spearman’s correlation coefficient was calculated to determine the correlation between SUVlbm Max and other variables, like tumor size and gender. Continuous variables were expressed as mean ± standard deviation (SD), and categorical variables as percentages. 

## 3. Results

The focal FDG uptake in the thyroid gland was observed in 126 out of 6900 patients, represented by 92 (73%) females and 34 (27%) males, with a mean age ± SD of 62 ± 13 years old.

The distribution of TIs in the thyroid gland was as follows: 61 located in the right lobe, 46 in the left lobe, 10 in the isthmus, and 9 were bilaterally located. The SUVlbm Max of these TIs ranged between 1.27 and 22.29, with a mean ± SD of 4.44 ± 3.68. 

Among these cases, 87 out of the 126 patients were US evaluated. Considering the US nodules criteria and the patients’ consent, only 29 of them were investigated using the FNAB procedure. 

The following report system was used: Thy1-nondiagnostic or unsatisfactory; Thy2-benign; Thy3-atypia of undetermined significance (AUS), or follicular lesion of undetermined significance (FLUS); Thy4-follicular neoplasm or suspicious for a follicular neoplasm; Thy5-suspicious for malignancy; and Thy6-malignant.

As specified by the Bethesda System for reporting thyroid cytopathology, benign lesions were reported (Thy2) in 16 cases (55.17%); despite the results, two patients chose to undergo a total thyroidectomy, and the benign diagnosis was confirmed in five cases (17.24%), where the cytopathology evaluation was unsatisfactory or undetermined (Thy1 and Thy3), and all of them were followed-up in time by laboratory tests and US. 

In eight cases (27.5%), the report was suggestive for malignancy after FNAB (Thy5 and Thy6). Seven patients gave their approval for a total thyroidectomy, and the malignancy was proven in six cases. In one case, the result was represented by nodular goiter. Figure 2 shows an increased FDG uptake incidentally found in a patient followed-up with FDG PET/CT for cervix cancer; the histology report after thyroidectomy revealed papillary thyroid carcinoma.

In two cases, a total thyroidectomy was performed without FNAB evaluation, and the pathology report confirmed the malignancy. 

The histological types were represented as follows: four cases of papillary thyroid carcinoma; two cases of follicular carcinoma; one thyroid lymphoma; and in three cases, metastatic lesions from melanoma, colon cancer, and oral cavity cancer.

The SUVlbm Max distribution is presented in Figure 3. 

The mean SUVlbm Max for malignant and benign nodules suggests higher values for malignant nodules, but the difference was not statistically significant (*p* = 0.08; Table 1). The mean ± SD SUVlbm Max was determined to be 12.1 ± 4.76 for metastatic lesions, and 4.97 ± 1.51 for thyroid carcinoma. The SUVlbm Max for the metastatic lesions was significantly higher, and the difference was statistically significant (*p* = 0.004).

Simultaneously, we arbitrarily established a SUVlbm Max cutoff value of four; the results are presented in Table 2. 

We calculated Spearman’s correlation coefficient to determinate the correlation between SUVlbm Max and the size of the tumor, resulting in a moderate positive correlation (*r* = 0.508). On the other hand, no correlation between SUVlbm Max and age (*r* = 0.02) was established, and there were no statistically significant differences in SUVlbm Max for the males and females (*p* = 0.73). 

## 4. Discussion

Over the past decade, thyroid carcinoma incidence has been increasing, with an important impact on life quality [14], and TIs diagnosed on FDG PET/CT represent a subject of considerable interest and research [15].

In our study of 6900 cases, focal TIs were identified in 1.82% (126/6900) of patients. This percentage is within the 0.2–8.9% range reported in the literature [16,17]. While Bertagna et al. [17] reported a TIs incidence of 2.5% for a group study of 147,505 patients, Hagenimana et al. [18] found that just 0.74% out of 40,914 patients presented TIs on FDG PET/CT reports.

The malignancy rate was found to be 32.25% (*n* = 10) if we take into consideration just those cases with a final histopathological report, these values are in range (8–64%) with other previous reported data [7,16,19,20,21,22]. Thyroid US was used to stratify the risk of malignancy for thyroid nodules and had a decisional role for performing FNAB. Even if 87 patients from our study underwent a US evaluation, just a small percentage of patients (*n* = 29; 24.6%) underwent a FNAB procedure, and two of them underwent a total thyroidectomy. If we take into consideration the entire study group (n = 126), we can report a 7.93% rate of malignancy for TIs (10/126). On the other side, 31% of cases were not investigated at all by US and FNAB, even if, according to the American Thyroid Association (ATA) guidelines, all thyroid nodules incidentally discovered on PET/CT and confirmed through US, having a dimension of >1 cm and a suspicious US criteria, should be FNAB evaluated [12]. 

Otherwise, if the number of patients who performed a FNAB would be higher, the malignancy rate would decrease; this idea was also discussed in a complex overview on 27 studies of Bertagna et al. [17].

The malignancy was diagnosed through FNAB in 8 out of 10 cases, so we can assume that FNAB was the primary diagnostic step in the management of this category of patients. Even though FNAB is a low-risk procedure, the small number of FNAB procedures represents a limitation of our study, which leads to an underestimation of the malignancy rate. We consider that this aspect is strongly associated with primary disease status. Analyzing our data, we noticed that the decision to start further investigations is taken very late, and depends on the medical state of the patient. If the primary tumor is stable, with a good prognosis, the patient is encouraged to perform further investigation regarding the incidental thyroid findings. For patients with aggressive tumors in advanced stages, additional thyroid investigations were not taken into consideration or were postponed, unless the TI would suggest a second primary malignancy more aggressive than the initial tumor. 

It is also worth mentioning that 55.17% (16 cases) of patients who performed a FNAB procedure had a benign result (Thy2), and they were just followed-up in time without it being necessary to perform surgery. There are studies that stated that the FNAB procedure performed for thyroid nodules with suspicious US features had benign results in 36–75% of cases, and so, an unnecessarily high number of thyroid surgeries were avoided [23,24]. This aspect highlights the importance of FNAB in the evaluation process of TIs.

The most frequent histological type of TIs diagnosed in our study group is represented by papillary thyroid carcinoma in four cases (40%), which is the most common histological type, with an excellent prognosis and high survival rate. The high percentage of papillary thyroid carcinoma was also revealed in other studies, in some of them, being present in 100% of diagnosed thyroid nodules [7,17,23,25,26,27].

It is known that the malignant lesions have higher SUV values than benign ones. It is still debated whether the SUV value could be considered a predictive parameter for malignancy or not; in our cohort of patients, we found that there is no statistically significant difference between SUVlbm Max mean (6.932 ± 4.39) for malignant Tis, and SUVlbm Max mean (4.22 ± 3.55) for benign TIs (*p* = 0.08). Even if we consider prospective studies or meta-analysis realized on a high number of patients, in some of them, the author affirms that there is no direct correlation between SUV and malignancy; others show the opposite. That is why a final and firm decision cannot be taken regarding the impact of the SUV value on diagnostic management for these patients. Some authors remark that the SUV values of malignant lesions overlap with those of benign lesions, especially in cases of Hurthle cell adenomas, because of the presence of a high number of mitochondria [28], and this was also a reason they could not establish a conclusion regarding a cutoff value for SUV. Based on our work, we concluded that a SUVlbm Max cutoff value >4 is significantly associated with malignancy (*p* = 0.0168), and 8 out of 10 malignant cases have a SUVlbm Max over 4. In a retrospective study, Wang et al. analyzed different SUV thresholds, and considered that a value of 3.3 has a high sensitivity, and, taking into consideration the risk of malignancy, it is better to over-investigate these cases than to under-investigate them [29].

Another important aspect of our work was that we found a moderate positive correlation between SUVlbm Max and tumor size, so the diameter of the lesion influences the FDG uptake. Measuring the SUV value in a small lesion can be challenging, because of the partial volume effect, which can cause an underestimation of the radiopharmaceutical uptake, so lower values of SUV are found [30,31]. On the other side, taking into consideration the spatial resolution of the PET/CT device, it is known that lesions smaller than 5 mm are hard to detect. In this situation, special attention for reporting PET/CT results and a detailed evaluation for TIs are required. 

The most common primary malignancies where TIs were identified were breast cancer, Hodgkin lymphoma, colon cancer, and malignant melanoma. There were no selection criteria for the patients, despite the potential association of thyroid cancer with other malignancies or possibilities of frequent metastases in the thyroid or other cancers, so this can be considered a potential selection bias. The fact that this study is a single-center study can also lead to selection bias. The malignancy rate might be different if the study could be extended to other centers and a high number of patients could be further investigated for TIs.

## 5. Conclusions

Focal thyroid FDG uptake was found in 1.82% of oncologic PET/CT scans, with a malignancy rate of 7.93%. In our study, we can conclude that TIs with higher SUVlbm Max values had a tendency to be malignant, while benign TIs had lower SUVlbm Max values, and a cutoff value of four could be taken into consideration when the risk of malignancy is evaluated. Even if just a small percentage of patients were diagnosed through FNAB, this method remains the primary diagnostic step from the therapeutic management of these patients, and has a high impact on the surgical decision.

Based on our experience, we consider that each case of the TIs discovered on FDG PET/CT should be separately analyzed. A better collaboration between the medical team and the patient will lead to the decision of performing further investigations, in order to avoid missing malignant thyroid tumors.

## Figures and Tables

**Figure 1 diagnostics-09-00092-f001:**
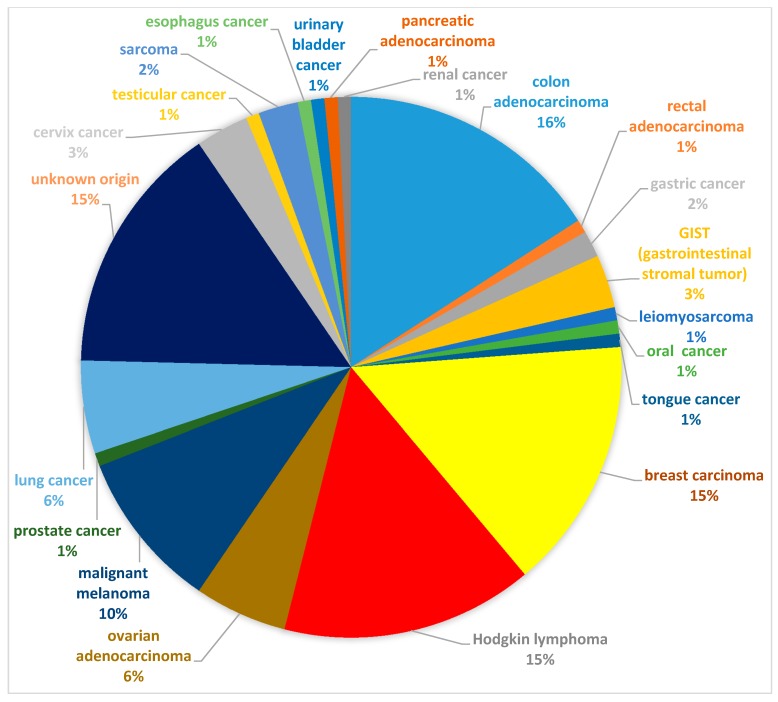
Types and distribution of primary malignant diseases.

**Figure 2 diagnostics-09-00092-f002:**
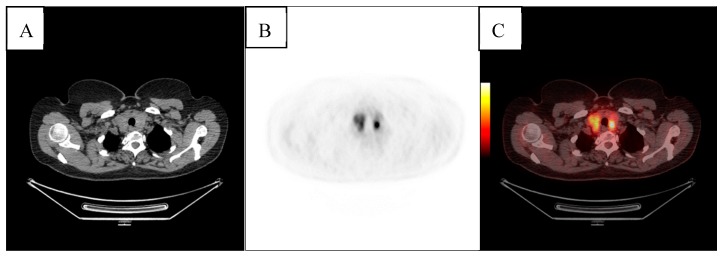
Axial section computed tomography (CT) (**A**), axial section positron emission tomography (PET) (**B**), and fusion (**C**) Fluorine-18 (F18)-18 fluoro-2-deoxy-d-glucose (FDG)-positron emission tomography/computed tomography (PET/CT) showing an increased FDG uptake in the multinodular thyroid gland of a patient followed-up with FDG PET/CT for cervix cancer; the histology report after thyroidectomy revealed papillary thyroid carcinoma incidentally identified in PET/CT.

**Figure 3 diagnostics-09-00092-f003:**
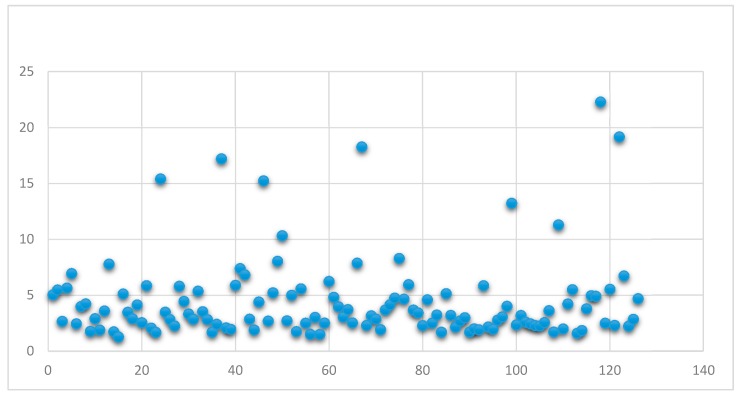
**SUVlbm** Max distribution.

**Table 1 diagnostics-09-00092-t001:** The standardized uptake value lean body mass (SUVlbm Max) mean for benign and malignant lesions. SD—standard deviation.

Parameter	Cancer/Metastasis	Benign
Mean ± SD SUVlbm Max	6.932 ± 4.39	4.22 ± 3.55
*p*	*p* = 0.08	

**Table 2 diagnostics-09-00092-t002:** SUVlbm Max cutoff value.

SUVlbm Max	Number of Patients	Patients with Malignancy (%)
SUVlbm Max <4	78	2 (2.56%)
SUVlbm Max >4	48	8 (16.6%)
*p*	0.0168	

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
