# Peer review of "Evaluation of Malignancy Risk in 18F-FDG PET/CT Thyroid Incidentalomas"

_diagnostics, 2019, doi:10.3390/diagnostics9030092_

Round 1

Reviewer 1 Report

Dear authors,

I think that this paper is very interesting and the topic is an open issue and not well-investigated.

The authors focused on the metabolic behaviour of thyroid incidentaloma at 18F-FDG PET/CT. Thyroid incidentaloma is a diagnostic challenge because it is not so easy to understand the nature (benign or malignant) of thyroid focal 18F-FDG uptake. Several papers in literature have tried to answer this question with different results.

The topic presented by authors and their analysis are fascinating both for nuclear medicine physicians and endocrinologist.

However, in the paper are present some points to improve and correct to let the paper more clear and complete.

After these improvement and corrections, I think that the article could be accepted.

INTRODUCTION

-          Line 37-41: I perfectly agree with the description of several imaging techniques (ultrasound, MRI, CT and PET) which potentially can discover thyroid incidentaloma, but considering nuclear medicine field not only FDG can recognize thyroid incidentaloma. Also with other radiotracers (PSMA, choline) thyroid incidental focal uptake are described (68Ga-PSMA PET thyroid incidentalomas. doi: 10.1007/s42000-019-00106-8.; F18-choline/C11-choline PET/CT thyroid incidentalomas. doi: 10.1007/s12020-019-01841-z.)

MATERIALS AND METHODS

-paragraph about statistical analysis lacks. Also descriptive analysis is a statistical analysis that has to be described.

- you have chosen SUVlbm, but what SUV value? Max? mean?  

- why not SUV body weight or body surface area? Explain it

RESULTS

- why 2 patients with Thyr2 (benign) performed total thyroidectomy? Please explain it... for goiter? For symptoms? For patient choice?

- the choice of 4 as cutoff for SUVlbm is an arbitrary choice that I don’t understand. Why 4? You could use the median value or use a ROC curve analysis to find the value with the best compromise between sensitivity and specificity. This is a crucial point to clarify.

- was there a difference between SUVlbm in primary thyroid carcinoma and metastatic lesion?

DISCUSSION

-When you explain the possible cause of relationship between tumor size and SUV you correctly cited partial volume effect. I’d add also the resolution power of the PET device; we know that lesion under 5 mm are not detectable with PET/CT

FIGURE 1

-Add % for every kind of neoplastic disease, so the image would be more clear.

FIGURE 2

-In figure 2 I’d add a image only with PET scan to show better the contrast between focal uptake and remaining thyroid tissue.

REFERENCES

- Searching in literature I’ve founded a recent review related to your topic that you have to include.

 Prevalence and clinical significance of focal incidental 18F-FDG uptake in different organs: an evidence-based summary. Tamburello, A., Treglia, G., Albano, D. et al. Clin Transl Imaging (2017) 5: 525. https://doi.org/10.1007/s40336-017-0253-8

Author Response

Dear Reviewer,

We are writing you regarding the manuscript ID Diagnostics-562562 that has been reviewed.

We are very grateful that our manuscript was taken into consideration and we would like to thank you also for the very kind recommendations.

We have performed the changes according to the attached comments, please find the response below. We uploaded the revised manuscript, with all the performed changes using "Track Changes", so they can be easily observed.

We also performed the English language spell check as you required.

Sincerely,

Maria-Iulia Larg

Institute of Oncology “Prof. Dr. Ion Chiricuta”

34-36 Republicii St.

400015, Cluj-Napoca

ROMANIA

Mobile: +40-0745869829

Response to Reviewer 1 Comments:

INTRODUCTION

-          Line 37-41: I fully agree with the description of several imaging techniques (ultrasound, MRI, CT and PET) which potentially can discover thyroid incidentaloma, but considering nuclear medicine field not only FDG can recognize thyroid incidentaloma. Also with other radiotracers (PSMA, choline) thyroid incidental focal uptake are described (68Ga-PSMA PET thyroid incidentalomas. doi: 10.1007/s42000-019-00106-8.; F18-choline/C11-choline PET/CT thyroid incidentalomas. doi: 10.1007/s12020-019-01841-z.)

 Thank you very much for the suggestion, we introduced a specific paragraph in the main document.

MATERIALS AND METHODS

-paragraph about statistical analysis lacks. Also descriptive analysis is a statistical analysis that has to be described.

We introduced in the main document the paragraph regarding statistical analysis.

- you have chosen SUVlbm, but what SUV value? Max? mean?  

We chose SUV lbm Max value and we wrote this in the main document.  

- why not SUV body weight or body surface area? Explain it

The PET/CT results were reported using SUV lbm because the standard of the PET/CT department, which is EARL accredited, imposed the use of SUV lbm which is modulated by glucose level.

RESULTS

- why 2 patients with Thyr2 (benign) performed total thyroidectomy? Please explain it... for goiter? For symptoms? For patient choice?

Unfortunately, in the clinical daily practice there is still a high number of total thyroidectomies performed for patient’s choice, which was also the situation in our case.  

- the choice of 4 as cutoff for SUVlbm is an arbitrary choice that I don’t understand. Why 4? You could use the median value or use a ROC curve analysis to find the value with the best compromise between sensitivity and specificity. This is a crucial point to clarify.

We have tested several values according with data from literature and we found that value of 4 was the lowest value with significance.

- was there a difference between SUVlbm in primary thyroid carcinoma and metastatic lesion?

Yes, we found that there was a statistically significant difference between two of them and we introduced a paragraph in the main document.

DISCUSSION

-When you explain the possible cause of relationship between tumor size and SUV you correctly cited partial volume effect. I’d add also the resolution power of the PET device; we know that lesion under 5 mm are not detectable with PET/CT

 Thank you for the idea, we put the paragraph in the main document.

FIGURE 1

-Add % for every kind of neoplastic disease, so the image would be more clear.

We introduced the % in the figure, it is clearer now.

FIGURE 2

-In figure 2 I’d add a image only with PET scan to show better the contrast between focal uptake and remaining thyroid tissue.

 Thank you for the suggestion, we included an axial image of PET.

REFERENCES

- Searching in literature I’ve founded a recent review related to your topic that you have to include.

 We included the review in references list, thank you for the suggestion.

Prevalence and clinical significance of focal incidental 18F-FDG uptake in different organs: an evidence-based summary. Tamburello, A., Treglia, G., Albano, D. et al. Clin Transl Imaging (2017) 5: 525. https://doi.org/10.1007/s40336-017-0253-8

Reviewer 2 Report

Interesting report about the frequency of off-target findings in the thyroid during FDG-PET and ist respective origin. Good patient number. The discussion should be amended by a paragraph "potential selection biases" mentioning that nutriation (especially iodine) can affect the pre-test ratio for the expected frequency of benign vs. malignant findings. Also the indication to perform FDG-PET presents an selection bias (e.g. renal cancer has a tendency for thyroid metastases but is also a poor indication for FDG) and reflects local habits.

I would also omit the conclusion that, FDG-PET would have an important role in TIs evaluation. In this work, FDG-PET was never performed under the indication TI evaluation but for staging of other cancers and off-target findings - and required further evaluation e.g. per FNAB.

Author Response

Dear Reviewer,

We are writing you regarding the manuscript ID Diagnostics-562562 that has been reviewed.

We are very grateful that our manuscript was taken into consideration and we would like to thank you also for the very kind recommendations.

We have performed the changes according to the attached comments, please find the response below. We uploaded the revised manuscript, with all the performed changes using "Track Changes", so they can be easily observed.

We also performed the English language spell check as you required.

Sincerely,

Maria-Iulia Larg

Institute of Oncology “Prof. Dr. Ion Chiricuta”

34-36 Republicii St.

400015, Cluj-Napoca

ROMANIA

Mobile: +40-0745869829

Response to Reviewer 2 Comments:

Interesting report about the frequency of off-target findings in the thyroid during FDG-PET and ist respective origin. Good patient number. The discussion should be amended by a paragraph "potential selection biases" mentioning that nutriation (especially iodine) can affect the pre-test ratio for the expected frequency of benign vs. malignant findings.

The patients from our study are from all parts of Romania representing 21 different counties. Nowadays, Romania has a national iodine fortification program since 2002. The general national specific data support the fact that the population of Romania is not with iodine deficiency; so considering these facts the selected population is assumed to be with normal iodine levels.    

Also the indication to perform FDG-PET presents an selection bias (e.g. renal cancer has a tendency for thyroid metastases but is also a poor indication for FDG) and reflects local habits.

There were no selection criteria of the patients despite the potential association of the primary tumor with other malignancies or possibilities of frequent metastases in thyroid of other cancers like renal cancer or malignant melanoma. We made a comment in the main document.

I would also omit the conclusion that, FDG-PET would have an important role in TIs evaluation. In this work, FDG-PET was never performed under the indication TI evaluation but for staging of other cancers and off-target findings - and required further evaluation e.g. per FNAB.

We took into consideration and we modified the conclusion accordingly.